# Changed Digital Technology Perceptions and Influencing Factors among Older Adults during the COVID-19 Pandemic

**DOI:** 10.3390/healthcare11152146

**Published:** 2023-07-27

**Authors:** Ok-Hee Cho, Junghee Cho

**Affiliations:** Department of Nursing, College of Nursing and Health, Kongju National University, Gongju 32588, Republic of Korea; ohcho@kongju.ac.kr

**Keywords:** COVID-19, digital healthcare services, digital technology, nationwide survey, older adults, perceived health

## Abstract

This correlational study aimed to identify factors that contribute to changes in perceptions of digital technology among older adults during the COVID-19 pandemic. This study utilized raw data from “The 2021 Report on the Digital Divide,” a nationwide survey conducted in South Korea. Data were collected from 1171 older adults (aged ≥ 65 years) from September to December 2021. Multiple regression analyses were performed to examine the factors influencing changes in the perception of digital technology. Over one-third of the participants reported positive changes in their perceptions of digital technology during the pandemic. Key factors included self-efficacy for digital devices (β = 0.35, *p* < 0.001), digital networking (β = 0.11, *p* < 0.001), accessibility to digital devices (β = 0.10, *p* = 0.002), and perceived health (β = 0.08, *p* = 0.003). The expansion of digital technology owing to the pandemic has served as a catalyst for changes in older adults’ perceptions. Healthcare providers and caregivers should consider digital technology perceptions and influencing factors when providing digital healthcare services. The results can be utilized to identify vulnerable older adults with negative perceptions of digital technology, thus minimizing disparities in access to digital healthcare services.

## 1. Introduction

Health-related digital technologies such as robots, the Internet of Things (IoT), artificial intelligence (AI), and Big Data have brought great changes to the medical industry, [1] and the COVID-19 pandemic has further accelerated and amplified the changes moving us toward a digital society. The prolonged pandemic has expanded non-contact culture and promoted the utilization of digital technology and diversification and normalization of digital healthcare services in the healthcare field [2]. However, recent studies have expressed concerns that the rapid expansion of digital technology during the pandemic has further widened pre-existing digital divides and could have implications for health inequalities within local communities [3,4]. Various contributing factors to vulnerabilities in digital technology adoption include old age [4], illiteracy or low educational attainment, low health and digital literacy [5], mental or cognitive impairments [6], mental health conditions such as anxiety or depression [7], physical health issues [5], low reliability of information [3], and institutionalized living arrangements [8].

Older adults are a vulnerable population in the digital divide owing to not only environmental limitations such as a lack of digital devices, restricted internet access, and low frequency of use before the pandemic [9,10] but also age-related characteristics such as declines in sensory and cognitive functions [11], lack of interest in digital devices, low confidence in digital device utilization, and difficulties in acquiring skills to use them [12]. Despite recognizing the usefulness of digital technology [13], older adults tend to have negative perceptions of unfamiliar digital technologies owing to their cautiousness toward new things [14]. Such negative perceptions and passive attitudes toward digital technology have made it more difficult for older adults than other age groups to acquire digital information on topics such as the latest updates on the pandemic, self-management after infection, and changes in strategies during the pandemic [15]. Consequently, older adults have faced challenges in utilizing digital technology for tasks such as setting and confirming vaccination appointments, attending online medical appointments, and making online payments, thereby making them unable to receive the benefits of digital technology [16].

The technology acceptance model proposes two levels of belief in behavior, perceived usefulness, and perceived ease of use that directly or indirectly affect users’ attitudes and intentions to use new technologies, services, and systems [17]. However, a web-based survey conducted by Van Deursen [18] involving 1733 Dutch participants reported that an older age was associated with a higher usage of information from the Internet, but older adults could not derive as many beneficial outcomes from using that information as the younger generation could. Further, both the usage and outcomes of internet communication were negatively correlated with age, indicating the importance of addressing the issues related to older adults’ perceptions and use of digital technology. 

As non-contact services expanded to prevent infection during the pandemic, the digital information gap caused older adults to face situations in which they had no choice but to opt-in for in-person services. This resulted in a paradoxical situation in which physically vulnerable older adults engaged in more physical interactions, exposing themselves to greater risks. Moreover, older adults used unidirectional media more than media capable of two-way communication or fast connections during the pandemic [19], suggesting that they could be more likely than other age groups to be unable to quickly and proactively access the necessary information [20]. 

Positive perceptions of digital technology are essential for obtaining and using beneficial information. Individuals with higher levels of awareness are more likely to utilize technology for easy access to digital healthcare services [21]. Internet/mobile-based health information resources using digital technology offer a wide range of content at relatively low costs [22] and provide individuals with opportunities to proactively acquire information and manage their health [23]. Accessibility, proficiency, and self-efficacy in using digital technology contribute to positive perceptions of its usability and usefulness, leading to increased usage and outcomes [3,4]. Further, older adults who recognize and experience the benefits of digital technology develop positive attitudes toward it [21], which further promotes the use of technology to reduce feelings of depression and loneliness and improve older adults’ overall quality of life [24].

The unprecedented experience of the pandemic is believed to have influenced perceptions of digital technology among older adults; however, studies on this topic remain scarce. Understanding the changes in older adults’ perceptions of digital technology can help enhance their usage intention and stimulate their proficiency and motivation for learning. Such understanding can serve as a catalyst for facilitating their adoption of digital technology. Further, identifying the factors influencing perception changes can help reinforce the use and adoption of digital solutions among older adults and establish strategies that promote positive outcomes. Therefore, this study was conducted using data from “The 2021 Report on the Digital Divide” to identify the factors that affected older adults’ perceptions of digital technology during the pandemic.

## 2. Research Design and Methods

### 2.1. Research Design

This was a correlational study. This study analyzed raw data from a nationwide survey, which has the advantage of being able to conduct empirical research on large-scale data. In addition, this method saves time and costs required for data collection, can obtain data for research subjects that are difficult to survey, such as the elderly, and has excellent methodological rigor, such as research design and measurement.

### 2.2. Study Sample and Data Collection

This study utilized raw data from “The 2021 Report on the Digital Divide,” a nationwide survey conducted annually by the National Information Society Agency (NIA) and the Ministry of Science and ICT in South Korea to provide fundamental data for policies aimed at reducing the digital information gap [25]. Data were acquired by following the data request procedure specified by the NIA and obtaining approval, after which raw data and a codebook were downloaded from the website. The survey collected data from 15,000 respondents who were selected through proportionate stratified sampling, targeting the general population aged 7 years or older residing in households across the country as of 1 August 2021. For this study, data from 1171 adults aged 65 years or older were analyzed. The respondents included in this study were those who currently used personal computers (desktops/laptops) and smart devices (e.g., smartphones and tablets). The survey method involved face-to-face interviews using a structured questionnaire, and the data collection period was September–December 2021.

### 2.3. Research Instruments

#### 2.3.1. Demographic Characteristics 

The demographic characteristics examined in this study included sex (male/female), age (65–74 years/75 years or older), economic activities (yes/no), education (≤elementary school/middle school/≥high school), monthly income (KRW < 1 million/KRW 1–2.99 million/KRW ≥ 3 million), household type (living alone/two or more), residential area (city/county), and perceived health (satisfied/dissatisfied).

#### 2.3.2. Digital Networking

The following items were used to measure digital networking: “I have used the Internet to maintain and develop relationships with people I already know” and “I have used the Internet to meet new people and communicate with them.” Participants were categorized as “yes” if they answered *occasionally* or *frequently* or “no” if they answered *rarely* or *never* to these two items.

#### 2.3.3. Accessibility to Digital Devices

Accessibility to digital devices was assessed using six items, including “ownership of a desktop computer, laptop, mobile phone, tablet, or smart accessories” and “availability of home Internet or a wired/wireless Internet connection.” Following the guidelines of the survey institution [25], the items were weighted and scored out of 100, with higher scores indicating greater accessibility to digital devices.

#### 2.3.4. Ability to Use Digital Devices

The seven items used to measure digital device proficiency included the “ability to perform basic activities using a personal computer/mobile device (basic system setup, wireless network configuration, file transfer to/from a computer, file sharing with others, installation and use of necessary applications, malware detection and treatment, document and data creation).” Following the guidelines of the survey institution [25], the items were weighted and scored out of 100, with higher scores indicating higher proficiency in using digital devices. 

#### 2.3.5. Self-Efficacy for Digital Devices

The items used to measure self-efficacy for digital devices were “I am confident in learning to use digital devices,” “I am confident in utilizing digital devices,” “I can quickly learn how to use digital devices,” and “I want to increase my use of digital devices.” Each of the four items was rated on a four-point scale (*strongly agree* = 4 points, *strongly disagree* = 1 point), resulting in total scores ranging between 4 and 16. Higher scores indicated higher levels of self-efficacy for digital devices.

#### 2.3.6. Changed Perceptions of Digital Technology

Changes in perceptions of internet/mobile-based digital technology during the pandemic were measured using three items: “The Internet and mobile devices have become more important in my life,” “I will be left behind in society if I lack the ability to use the Internet and mobile devices,” and “I wish I had more opportunities to learn about the Internet and mobile devices.” Each item was rated on a five-point scale (*strongly agree* = 5 points, *strongly disagree* = 1 point), resulting in total scores ranging between 3 and 15. Higher scores indicated greater positive change in perceptions of digital technology.

### 2.4. Ethical Considerations

This study received approval for review exemption (no. KNU_IRB_2022-096) from the Institutional Review Board at K University.

### 2.5. Data Analysis

SPSS/WIN 26.0 software (IBM, Armonk, NY, USA) was used for data analysis. Descriptive statistics were used to examine participants’ general characteristics and the levels of the research variables. Differences in changed digital technology perceptions were assessed using independent t-tests, one-way analyses of variance (ANOVAs), and Scheffé’s post hoc test. Cases in which the assumption of homogeneity of variance was not met during the ANOVA were analyzed using Welch’s test and the Games–Howell post hoc test. The correlations between research variables were analyzed using Pearson’s correlation coefficient, and the factors influencing changed perceptions of digital technology were analyzed using multiple regression analyses.

## 3. Results

### 3.1. Demographic Characteristics and Levels of Research Variables

Participants’ general characteristics are shown in Table 1. Participants had an average digital accessibility score of 76.30 ± 23.70 out of 100, an average digital device proficiency score of 20.75 ± 27.81 out of 100, and an average digital device self-efficacy score of 1.97 ± 0.74 out of 100. The average score for changed perceptions of digital technology was 3.09 ± 0.84 out of 5. More than one-third (38.6%; 43.3% of those aged 65–74 years and 25.1% of those aged ≥ 75 years) of participants agreed (*agree* or *strongly agree*) that “the Internet and mobile technology have become more important in their lives” during the pandemic. Further, 35.6% (39.1% of those aged 65–74 years and 25.7% of those aged ≥ 75 years) agreed that “they will be left behind in society if they lack Internet and mobile technology skills”, and 33.2% (38.0% of those aged 65–74 years and 19.4% of those aged ≥ 75 years) agreed that “they wish they had more opportunities to learn Internet and mobile technology”.

### 3.2. Correlations between Digital Technology Perceptions and Research Variables 

Changed perceptions of digital technology showed significant differences according to sex (t = 4.58, *p* < 0.001), age (F = 6.77, *p* < 0.001), economic activities (F = −4.68, *p* < 0.001), education (F = 75.91, *p* < 0.001), monthly income (F = 43.99, *p* < 0.001), household type (t = 4.05, *p* < 0.001), residential area (t = 2.45, *p* = 0.014), perceived health (t = 9.63, *p* < 0.001), and digital networking (t = 13.34, *p* < 0.001; Table 1). Changed perceptions of digital technology showed positive correlations with accessibility to digital devices (r = 0.387, *p* < 0.001), ability to use digital devices (r = 0.370, *p* < 0.001), and self-efficacy for digital devices (r = 0.511, *p* < 0.001; Table 2).

### 3.3. Influencing Factors for Changed Digital Technology Perceptions

A multiple regression analysis was performed to investigate influencing factors and their impact on changed perceptions of digital technology during the COVID-19 pandemic. Variables that showed significant associations in the univariate analysis with the dependent variable (i.e., changed perceptions of digital technology) were used as independent variables in the regression analysis. These included sex, age, economic activities, education, monthly income, household type, residential area, perceived health, and digital networking (categorical variables were dummy-coded), as well as accessibility to digital devices, ability to use digital devices, and self-efficacy for digital devices. The derived regression model was significant (F = 38.15, *p* < 0.001) and explained 30.8% of the variance in changed perceptions of digital technology. Factors that showed a significant effect included self-efficacy for digital devices (β = 0.35, *p* < 0.001), digital networking (β = 0.11, *p* < 0.001), accessibility to digital devices (β = 0.10, *p* = 0.002), and perceived health (β = 0.08, *p* = 0.003; Table 3).

## 4. Discussion

This study utilized large-scale data from “The 2021 Report on the Digital Divide,” a nationwide survey in South Korea, to identify changed perceptions of digital technology among older adults during the pandemic and the influencing factors. Over one-third of the participants agreed that their perceptions of digital technology had changed. This is lower than the reported rates of 64.9–73.7% of the general population in South Korea and of the vulnerable groups such as individuals with disabilities, individuals with low-income, and farmers and fishers [25]. Notably, over 40% of older adults aged 65–74 and over 20% of those aged 75 or older experienced positive changes in their perceptions of digital technology during the pandemic. This suggests that a differentiated strategy targeting the young–old and old–old populations could be effective in promoting the adoption of digital technology among older adults. Further, it signifies the importance of addressing the ongoing educational needs and enthusiasm for digital technology among approximately one-fifth of the older adult population.

Several influencing factors led to changes in digital technology perceptions: self-efficacy for digital devices, digital networking, accessibility to digital devices, and perceived health. First, self-efficacy for digital devices was the most significant factor influencing perception changes. This finding aligns with previous studies in South Korea reporting that higher self-efficacy among older adults is associated with a greater intention to use digital health devices [15] and digital device adoption [26]. Self-efficacy is essential in all stages of life, and particularly in old age, as it is associated with beliefs, emotions, and behaviors and influences self-esteem and autonomy [27]. Self-efficacy for digital devices, based on beliefs in one’s ability to perform information technology tasks [28], enhances older adults’ confidence in their own capabilities, increases awareness of digital devices and information, and promotes the utilization of technology. It has positive effects on reducing anxiety and depression [24], improving fitness, raising awareness of falls, and recognizing the importance of physical activity [29,30]. These findings support the possibility that self-efficacy for digital devices goes beyond the subjective confidence in using digital devices and is associated with an attitude of actively utilizing technology.

From a different perspective, participants notably showed a relatively low level of self-efficacy for digital devices, with an average score of 1.97 out of 4. Although not significant as an influencing factor in the final model, the level of ability to use digital devices was also very low, with an average score of 20.75 out of 100. There is a need for diversified digital literacy education to be provided to older adults in local community centers and public health centers. To enhance perceptions of digital technology, effective programs that incorporate self-efficacy and learning elements centered around usability need to be developed.

Second, older adults who engaged in digital networking experienced greater changes in their perceptions of digital technology after the pandemic compared to those who did not. Participating in digital networking provides a higher likelihood of initiating meaningful connections with friends and family [31] and allows individuals to share news and experiences with people they cannot meet in-person, share resources, and contact institutions including healthcare organizations [32]. Sims et al. [31] stated that older adults primarily use digital technology for social purposes such as maintaining relationships with friends and family, rather than for informational purposes. Shapira et al. [33] reported that social networking contributed to reducing loneliness and depression among older adults during the pandemic. The circumstances of the COVID-19 pandemic encouraged older adults to engage in more digital networking to maintain social connections, and experiencing the usefulness of digital technology in mitigating social isolation and depression likely had a positive impact on their perceptions of digital technology. 

Third, the level of accessibility to digital devices was positively correlated with the extent of older adults’ changes in their perceptions of digital technology. Digital accessibility refers to the ability to access essential hardware, software, and internet services related to digital technology utilization [34]. Low digital accessibility among older adults can be an obstacle in providing remote healthcare services. Van Deursen [18] reported that, during the pandemic, general adults who were users of various types of digital devices, with high material access and skill access, saw more benefits and engaged in more digital information use. However, older adults were not able to obtain as many benefits as general adults were able to despite the increase in digital information use. Runfola et al. [35] provided digital devices to discharged patients and conducted teleconsultations at home; however, only 58% of the patients participated. Many studies have highlighted the need for education to enhance older adults’ utilization skills and literacy for digital device ownership to translate into actual usage [36,37]. Nevertheless, older adults must first change their perceptions to willingly embrace and acquire technological capabilities.

Finally, older adults who were satisfied with their health had greater changes in their perceptions of digital technology compared to those who were dissatisfied. Perceived health is a contributing factor in health-related decision-making [38], and the results indirectly suggest that older adults with higher health satisfaction also have greater intentions to adopt digital technology [31]. Repetitive use of digital devices driven by necessity and the pursuit of health information during the pandemic could have influenced positive perceptions of digital technology. However, issues regarding the reliability and validity of digital information are still under debate [39], and customized digital health information infrastructure should be improved for older adults with frailty and complex chronic diseases and who require management of multiple medications, in addition to developing the digital capabilities of medical staff.

This study had several limitations. First, this study selected similar attributes to the examined concepts to reflect the attributes of variables because panel data were available that longitudinally measured perceptions of digital technology and related variables. However, follow-up studies should determine the factors influencing direct changes by longitudinally measuring the process of variables. Second, the study did not control for some demographic variables that could affect perceptions of digital technology, and social variables such as social capital and assistance were not sufficiently reflected. In follow-up studies, a model including social factors such as retirement that can result in decreasing social contact should be verified considering the characteristics of old age. Third, the responses in this study may have been overestimated because it did not include older adults who did not use digital devices or were hospitalized or admitted to community facilities and alienated from digital information. Thus, the findings may not be generalizable to all older adults. Further, the measurements were constructed based on respondents’ answers, which could limit the accuracy. Fourth, future qualitative research could help provide an in-depth understanding of the acceptance factors and obstacles to digital technology use among older adults, and research is needed on expanded models, including perceptions of the importance, need, and educational needs of digital technology in the path of technology acceptance models. Fifth, this study’s results showed a significant impact on older adults’ perceptions of digital technology; therefore, experimental studies should be developed and applied to empirically verify these findings. Sixth, aspects such as availability and cost as conditional variables related to technology adoption are being reviewed [40]. In previous studies using the technology acceptance model, the burden of cost acted as a meaningful variable in digital technology adoption [17]. However, this study did not investigate how much participants paid for digital technology or the diversity of their digital technology resources. Therefore, follow-up studies should verify the correlations according to the economic situation or state of technological resources of older adults to increase the current understanding of changes in how older adults perceive digital technology. 

Despite these limitations, this study aimed to identify ways to strengthen older adults’ adaptability to digital technology by identifying variables that can affect positive perceptions of digital technology in this population. The findings of this study can help bridge the digital divide by promoting digital technology use through improving the current understanding of older adults’ perceptions of various emerging digital technologies and providing basic data on how to deliver digital-based healthcare services for older adults.

## 5. Conclusions

Over one-third of the 1171 older adults in this study responded that their perceptions of digital technology had changed during the pandemic. The factors influencing these changes included self-efficacy in digital technology, digital networking, accessibility of digital devices, and perceived health. Understanding the realities and influencing factors of changes in digital technology perceptions among older adults can provide a basis for digital content development and education that reinforces factors with a positive influence. The findings can also be utilized to develop digital health systems and formulate healthcare policies for healthcare providers and caregivers. The results will be helpful in identifying vulnerable older adults with negative perceptions of digital technology to minimize the disparities in access to digital healthcare services and facilitate equitable benefits in a society with rapidly accelerating digitalization following the pandemic.

## Figures and Tables

**Table 1 healthcare-11-02146-t001:** Differences in changed digital technology perceptions according to demographic characteristics (N = 1171).

Characteristics	Categories	Total	Digital Technology Perceptions	
		n (%)	Mean ± SD	t or F (*p*/Post hoc)
Sex	Male	509 (43.5)	3.22 ± 0.78	4.58 (<0.001)
	Female	662 (56.5)	2.99 ± 0.87	
Age (years)		71.9 ± 5.3 *		
	65∼74	872 (74.5)	3.19 ± 0.81	6.77 (<0.001)
	≥75	299 (25.5)	2.81 ± 0.86	
Economic activities	Yes	430 (36.7)	3.24 ± 0.79	4.68 (<0.001)
	No	741 (63.3)	3.00 ± 0.86	
Education	≤Elementary school ^a^	365 (31.2)	2.70 ± 0.88	75.91 (<0.001)
	Middle school ^b^	378 (32.3)	3.12 ± 0.80	a < b < c
	≥High school ^c^	427 (36.5)	3.40 ± 0.70	
Monthly household	<100 ^a^	188 (16.0)	2.66 ± 0.87	43.99 (<0.001)
Income (KRW 10,000)	100∼299 ^b^	643 (54.9)	3.07 ± 0.82	a < b < c
	≥300 ^c^	340 (29.1)	3.36 ± 0.76	
Household type	Living alone	236 (20.1)	2.89 ± 0.85	4.05 (<0.001)
	Two or more	935 (79.9)	3.14 ± 0.83	
Residential area unit	City	1059 (90.5)	3.11 ± 0.84	2.45 (0.014)
	Country	112 (9.5)	2.90 ± 0.83	
Perceived health status	Dissatisfied	553 (47.3)	2.85 ± 0.83	9.63 (<0.001)
	Satisfied	617 (52.7)	3.31 ± 0.79	
Digitally leveraged	Yes	430 (36.7)	3.48 ± 0.70	13.34 (<0.001)
networking	No	741 (63.3)	2.87 ± 0.83	

SD = Standard deviation; * = Mean ± SD; ^a^, ^b^, ^c^ = Post hoc by Scheffé test or Games-Howell test.

**Table 2 healthcare-11-02146-t002:** Correlations between research variables (N = 1171).

Variables	M ± SD	Pearson’s Correlation Coefficient *			
		1	2	3	4
1. Accessibility to Digital Information	76.30 ± 23.70				
2. Ability to Use Digital Devices	20.75 ± 27.81	0.546 (<0.001)			
3. Self-Efficacy for Digital Devices	1.97 ± 0.74	0.449 (<0.001)	0.478 (<0.001)		
4. Changed Digital Technology Perceptions	3.09 ± 0.84	0.387 (<0.001)	0.370 (<0.001)	0.511 (<0.001)	

* Values are r (*p*).

**Table 3 healthcare-11-02146-t003:** Influencing factors for changed digital technology perceptions (N = 1171).

Variables		B	SE	Β	T	*p*	95% CI	
							Lower	Upper
(Constant)		1.76	0.11		16.14	<0.001	1.55	1.98
Sex (ref: Female)	Male	0.02	0.05	0.01	0.40	0.693	−0.08	0.11
Age (years) (ref: ≥75)	65∼74	0.02	0.05	0.01	0.44	0.659	−0.08	0.12
Economic activities (ref: No)		−0.03	0.05	−0.01	−0.51	0.612	−0.12	0.07
Education (ref: ≤Elementary school)	Middle school	0.06	0.06	0.03	1.08	0.282	−0.05	0.17
	≥High school	0.06	0.06	0.03	0.94	0.346	−0.07	0.19
Monthly income (KRW 10,000)	100∼299	0.08	0.07	0.05	1.17	0.241	−0.05	0.21
(ref: ≤100)	≥300	0.09	0.09	0.05	1.01	0.315	−0.08	0.25
Household type (ref: Living alone)	Two or more	−0.01	0.06	−0.01	−0.24	0.811	−0.13	0.10
Residential area unit (ref: Country)	City	−0.02	0.07	−0.01	−0.32	0.746	−0.16	0.12
Perceived health status (ref: Dissatisfied)	Satisfied	0.13	0.05	0.08	2.96	0.003	0.05	0.22
Digitally leveraged networking (ref: No)	Yes	0.19	0.05	0.11	3.73	<0.001	0.09	0.29
Accessibility to digital information		0.01	0.01	0.10	3.09	0.002	0.01	0.01
Ability to use digital devices		0.01	0.01	0.06	1.71	0.088	0.01	0.01
Self-efficacy for digital devices		0.39	0.04	0.35	11.30	<0.001	0.33	0.46
R^2^		0.316						
Adjusted R^2^		0.308						
F(*p*)		38.15(<0.001)						

CI = confidence interval; ref. = reference.

## Data Availability

Not applicable.

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
