# Peer review of "Changed Digital Technology Perceptions and Influencing Factors among Older Adults during the COVID-19 Pandemic"

_healthcare, 2023, doi:10.3390/healthcare11152146_

Round 1

Reviewer 1 Report

The followings are my observations and suggestions while reviewing the paper,

1) Research contribution is very low, Author must describe their original research contribution. Describe research motivation and how they address the challenges in that regard.

2) Observational methodologies must be addressed with experiment with all possible situations. The methodologies needs a lot of improvements and recent improvements in this field need to cover in their research.

3) Is there any programing implementation, if yes show the GitHub link. If no, then what are the experiment you have done clearly describe experiment arrangement, how you perform the experiment and compare result with some benchmark.

Author Response

Response to Reviewer 1 Comments

Manuscript ID: healthcare-2500400

*Title: Changed digital technology perceptions and influencing fac-tors among older adults during the COVID-19 pandemic

We appreciate the time and effort that you and the reviewer have dedicated to providing your valuable feedback on my manuscript. We are grateful to the reviewer for your insightful comments on my paper. We have been able to incorporate changes to reflect most of the suggestions provided by the reviewer. We have highlighted the changes within the manuscript.

Here is a point-by-point response to the reviewer’ comments and concerns.

1) Research contribution is very low, Author must describe their original research contribution. Describe research motivation and how they address the challenges in that regard.

Response: PAGE 8, LINE 313-318. As suggested by the reviewer, we have supplemented the contents of the discussion about the contribution and utilization of the study.

2) Observational methodologies must be addressed with experiment with all possible situations. The methodologies needs a lot of improvements and recent improvements in this field need to cover in their research.

Response: PAGE 8, LINE 286-312. This study is a secondary analysis study of data from a nationwide survey in South Korea, and the methodological limitations and areas for improvement of this study are additionally described in the 'Discussion' section.

3) Is there any programing implementation, if yes show the GitHub link. If no, then what are the experiment you have done clearly describe experiment arrangement, how you perform the experiment and compare result with some benchmark.

Response: PAGE 8, LINE 286-312. This study is a cross-sectional correlation study, not a programming implementation or experimentation study. Based on the reviewer's suggestion, we additionally suggested that an experimental study could be conducted based on the results of this study.

Reviewer 2 Report

The study touches a relevant topic, however, the manuscript does not satisfactorily elaborate on a theoretical foundation and framework. For example, digital literacy and digital technologies are not further concretized and specified in terms of concrete technologies and internet/mobile-based technologies and applications. Comparable studies exist concerning IT literacy. ICT literacy is a subject addressed by professioal associations and is in focus on ongoing initiatives. The authors do not satisfactorily link to existing initiatives concerning ICT skills. I wondered why the survey participants were not asked about what concrete technologies they use, how much financial resources they have and spent for digital technologies. The author do not argue why the analyzed data and data sample appears representative in relation to the overall population. The section discussion of the paper does discuss previous studies, however, I thought this should be moved rather to the state-of-the-art analysis at the start, which is not suffciently well elaborated and detailed. The author do not argue why the applied empirical analysis is adequate to deliver required outcomes and results. Are there probably alternatives, why is this approach from the authors' perspective best fitting? Hence, from my point of view, the empirical research misses a conceptual base and theoretical foundation.

Quality of English is overall good, no significant shortcomings could be identified.

Author Response

Response to Reviewer 2 Comments

Manuscript ID: healthcare-2500400

*Title: Changed digital technology perceptions and influencing fac-tors among older adults during the COVID-19 pandemic

We appreciate the time and effort that you and the reviewer have dedicated to providing your valuable feedback on my manuscript. We are grateful to the reviewer for your insightful comments on my paper. We have been able to incorporate changes to reflect most of the suggestions provided by the reviewer. We have highlighted the changes within the manuscript.

Here is a point-by-point response to the reviewer’ comments and concerns.

1) The study touches a relevant topic, however, the manuscript does not satisfactorily elaborate on a theoretical foundation and framework. For example, digital literacy and digital technologies are not further concretized and specified in terms of concrete technologies and internet/mobile-based technologies and applications. Comparable studies exist concerning IT literacy. ICT literacy is a subject addressed by professioal associations and is in focus on ongoing initiatives. The authors do not satisfactorily link to existing initiatives concerning ICT skills. I wondered why the survey participants were not asked about what concrete technologies they use, how much financial resources they have and spent for digital technologies.

Response:

- PAGE 1, LINE 28-31. As suggested by the reviewers, we have added to the ‘Introduction’ section about the digital technologies currently in use.

- PAGE 3, LINE 107-108. In addition, the digital devices specifically used by the participants were additionally described in the ‘Method’ section.

- PAGE 8, LINE 286-312. We did not investigate how much the participants in this study paid for digital technology or how diverse digital technology resources they had. This is described as a limitation in the ‘Discussion’ section.

2) The author do not argue why the analyzed data and data sample appears representative in relation to the overall population. The section discussion of the paper does discuss previous studies, however, I thought this should be moved rather to the state-of-the-art analysis at the start, which is not suffciently well elaborated and detailed. The author do not argue why the applied empirical analysis is adequate to deliver required outcomes and results. Are there probably alternatives, why is this approach from the authors' perspective best fitting? Hence, from my point of view, the empirical research misses a conceptual base and theoretical foundation.

Response:

- PAGE 8, LINE 286-312. We describe limitations on the representativeness of our data sample in the ‘Discussion’ section.

- PAGE 2, LINE 56-58. In addition, the theoretical framework of the study was supplemented in the ‘Introduction’ section.

- PAGE 8, LINE 302-306. The researcher's opinion on the empirical experimental research that can be conducted based on the results of this study in the future is described in the 'Discussion' section.

Round 2

Reviewer 1 Report

This correlational study improved a lot after passing through the review process. Currently , I do not have any more objections as it is just a correlational study.

Author Response

Response to Reviewer 1 Comments

Manuscript ID: healthcare-2500400

*Title: Changed digital technology perceptions and influencing fac-tors among older adults during the COVID-19 pandemic

We appreciate the time and effort that you and the reviewer have dedicated to providing your valuable feedback on my manuscript.

Here is a point-by-point response to the reviewer’ comments and concerns.

1) This correlational study improved a lot after passing through the review process. Currently , I do not have any more objections as it is just a correlational study.

Response: Thank you for your great comment. We are grateful to the reviewer for your insightful comments on my paper.

Reviewer 2 Report

The authors have overall well responded to the reviewers comments. The theoretical foundation is now acceptable in relation to digital technologies. It is now clearer that the authors refer to a specific model and available survey data from a report. Digital literacy and skills are not addressed but this is from my point of view acceptable. I would recommend to argue why the applied empirical analysis is most appropriate and the authors should add a brief introductory paragraph accordinly in section 2 "Materials and Methods". The header should be renamed into "Research Design and Methods". Why is the applied approach and design fitting to solve the problem at hand.  Are there alternatives? Please argue here, so that the readers are able to understand your decisions. All other parts are acceptable now.

English language is appropriate, the authors should just do some minor checking and final review.

Author Response

Response to Reviewer 2 Comments

Manuscript ID: healthcare-2500400

*Title: Changed digital technology perceptions and influencing fac-tors among older adults during the COVID-19 pandemic

We appreciate the time and effort that you and the reviewer have dedicated to providing your valuable feedback on my manuscript. We are grateful to the reviewer for your insightful comments on my paper. We have been able to incorporate changes to reflect most of the suggestions provided by the reviewer. We have highlighted the changes within the manuscript.

Here is a point-by-point response to the reviewer’ comments and concerns.

1) The authors have overall well responded to the reviewers comments. The theoretical foundation is now acceptable in relation to digital technologies. It is now clearer that the authors refer to a specific model and available survey data from a report. Digital literacy and skills are not addressed but this is from my point of view acceptable. I would recommend to argue why the applied empirical analysis is most appropriate and the authors should add a brief introductory paragraph accordinly in section 2 "Materials and Methods". The header should be renamed into "Research Design and Methods". Why is the applied approach and design fitting to solve the problem at hand.  Are there alternatives? Please argue here, so that the readers are able to understand your decisions. All other parts are acceptable now.

Response:

- PAGE 2-3, LINE 94-100; PAGE 8, LINE 290-294. Thank you for your great comment. Revisions have been made according to your comments.
